# Simulation of Gauged and Ungauged Streamflow of Coastal Catchments across Australia

Mohammed Abdul Bari [1,*], Urooj Khan [2], Gnanathikkam Emmanuel Amirthanathan [3], Mayank Tuteja [4] and Richard Mark Laugesen [2]

1   Bureau of Meteorology, 1 Ord Street, West Perth, WA 6005, Australia
2   Bureau of Meteorology, The Treasury Building, Parkes Place West, Canberra, ACT 2600, Australia; urooj.khan@bom.gov.au (U.K.); richard.laugesen@bom.gov.au (R.M.L.)
3   Bureau of Meteorology, 700 Collins Street, Docklands, VIC 3008, Australia; gnanathikkam.amirthanathan@bom.gov.au
4   Finity Consulting, Level 7, 68 Harrington Street, The Rocks, NSW 2000, Australia; mtuteja@iinet.net.au
*   Correspondence: mohammed.bari@bom.gov.au

**Abstract:** Australia is a unique continent, surrounded by the ocean, and the majority of its catchments flow to the coast. Some of these catchments are gauged and others are ungauged. There are 405 gauged catchments covering 2,549,000 km$^2$ across the coastal regions of 12 drainage divisions in Australia, whereas there are 771 catchments conceptualised as ungauged covering additional 835,000 km$^2$. The spatial and temporal distribution of mean annual rainfall and potential evaporation (PET) vary significantly from one drainage division to another. We developed a continuous daily streamflow time series of all gauged and ungauged catchments from 1993 onwards. We applied the daily GR4J lumped conceptual model to these catchments. The performance of gauged catchments was analysed through (i) visual inspection of daily hydrographs, flow duration curves, and daily scatter plots; and (ii) performance metrics, including NSE and PBias. Based on the NSE and PBias, performance ratings of 80% and 96% of the models, respectively, were found to be 'good'. There was no relationship found between the catchment area and the model performance. The ungauged catchments were divided into four categories based on distance from potential donor catchments, where observed data are available for GR4J model calibration, and Köppen climate zone. The total ungauged catchments represent 24.7% of the total drainage division areas. The streamflow from ungauged catchments was estimated using the GR4J model based on the parameters of their donor catchments. Overall, runoff ratios from ungauged catchments were found to be higher compared to their donor-gauged catchments, likely driven by their higher rainfall and less PET. This tendency was particularly evident in two drainage divisions—the Carpentaria Coast (CC) and the Tanami–Timor Sea Coast (TTS)—where ungauged areas comprised 51% and 43%, respectively. The mean gauged annual streamflow varied significantly across drainage divisions—230 gigalitres (GL) from the South Australian Gulf (SAG) to 146,150 GL in TTS. The streamflow from all ungauged catchments was estimated at 232,200 GL per year. Overall, the average streamflow from all drainage divisions, including gauged and ungauged areas, across the coastal regions of Australia was estimated at 419,950 GL per year. This nationwide estimate of streamflow dataset could potentially enhance our understanding of coastal processes and lead to improvements in marine modelling systems and tools.

**Keywords:** GR4J model; coastal discharge estimates; ungauged catchments; streamflow simulations; inverse distance weighting; parameter transfer; Australia

## 1. Introduction

Water is an integral part of life and impacts directly on most aspects of human life and the environment. Catchments that supply this water integrate changes due to human activities and natural processes. Understanding river discharge is an important factor underpinning water management decisions [1]. Streamflow gauges are the principal

means of data collection and have been used for centuries. With the Nilometer as a prime example, stream discharge records are vitally important, and without these records, we cannot understand, observe, and manage our hydrologic systems for human development. Streamflow gauges are the most accurately measured component of the hydrological cycle [2].

As the streamflow measurement gauges can be built only at the finite locations in the stream network, they can only provide limited information in the space–time continuum [2]. The establishment and operations of the streamflow measurement gauges are costly; therefore, their location and operations largely depend upon national or local interests or funding from particular projects [3]. Even if the resources are available, it is not practically possible to build and operate flow measurement gauging stations at every possible location in the stream network. As a result, streamflow is only monitored in a small fraction of rivers in the world and most catchments remain completely ungauged [4–6]. This is a common problem prevalent in developed and developing nations, for example, the USA [3], the UK [7], Canada (Environment and Climate Change Canada: https://wateroffice.ec.gc.ca/mainmenu/historical_data_index_e.html, accessed on 2 November 2023), Asia [8], and Africa [9]. In Australia, Its Indigenous peoples have over 65,000 years of connection and understanding of water, and the value of water is central to their culture [10]. However, streamflow monitoring with gauges formally started as early as 1865 and expanded continuously till 1965. Since then, the monitoring network slightly declined [11]. Most of these streamflow measurement gauging stations are located in high-value water resource catchments, mainly in the coastal regions of Australia.

Australia is a marine nation and uniquely placed on the planet. Its marine state surrounds the entire continent, covers about 14 million $km^2$, and has a strong impact on terrestrial climates [12]. Its mainland coastline is approximately 38,910 km long [13] and has the third largest marine jurisdiction in the world. It is diverse and ranges from the tropics to the sub-Antarctic. Marine industries currently contribute over AUD 47 billion to Australia's economy [12]. The Australian Government's National Marine Science Plan 2015–2025 has highlighted challenges and emphasises the need to develop and refine decision-support tools that translate knowledge and data into useful information for effective decision-making in relation to these challenges. It also identifies the need for a coordinated national marine environment and socioeconomic modelling system. In response to these challenges, the Integrated Marine Observing System (IMOS) was established, and a large number of marine modelling systems were developed for research and implementation of the strategy [14,15]. For the further enhancement, accuracy, and efficient operations of these modelling systems, a nationwide quality-controlled stream discharge dataset of the major rivers flowing to the coastal regions of Australia will be highly beneficial as it could be taken as dynamic input to these systems. The nationwide coastal streamflow dataset will also be useful for ocean climate science research, model development, retrospective analyses, nowcasting, and forecasting.

Most of the river systems in Australia discharge into the coastal regions. There are only a limited number of gauging stations recording the discharge, and most of the rivers are ungauged (Figure 1). Without a consistent and comprehensive nationwide record of streamflow dataset, we are unable to improve our understanding of coastal processes and improve our marine modelling systems and tools. Therefore, the extension of the existing streamflow records and estimation of ungauged streamflow is vital for creating a complete nationwide dataset. At present, the dataset is not available, as no previous research has been undertaken to create this dataset. There are different procedures for estimating ungauged streamflow, as detailed in Section 4.3. In this study, we used a spatial proximity approach, which has not been previously attempted, with the following key objectives:

- Apply GR4J [16] daily rainfall–runoff models at all the coastal gauged catchments and evaluate their performance;
- Identify, cluster, and classify ungauged catchments into different categories;
- Transfer and apply GR4J models to all ungauged catchments and assess performance;

- Estimate daily and annual streamflow and create a nationwide coastal streamflow dataset for all gauged and ungauged catchments.

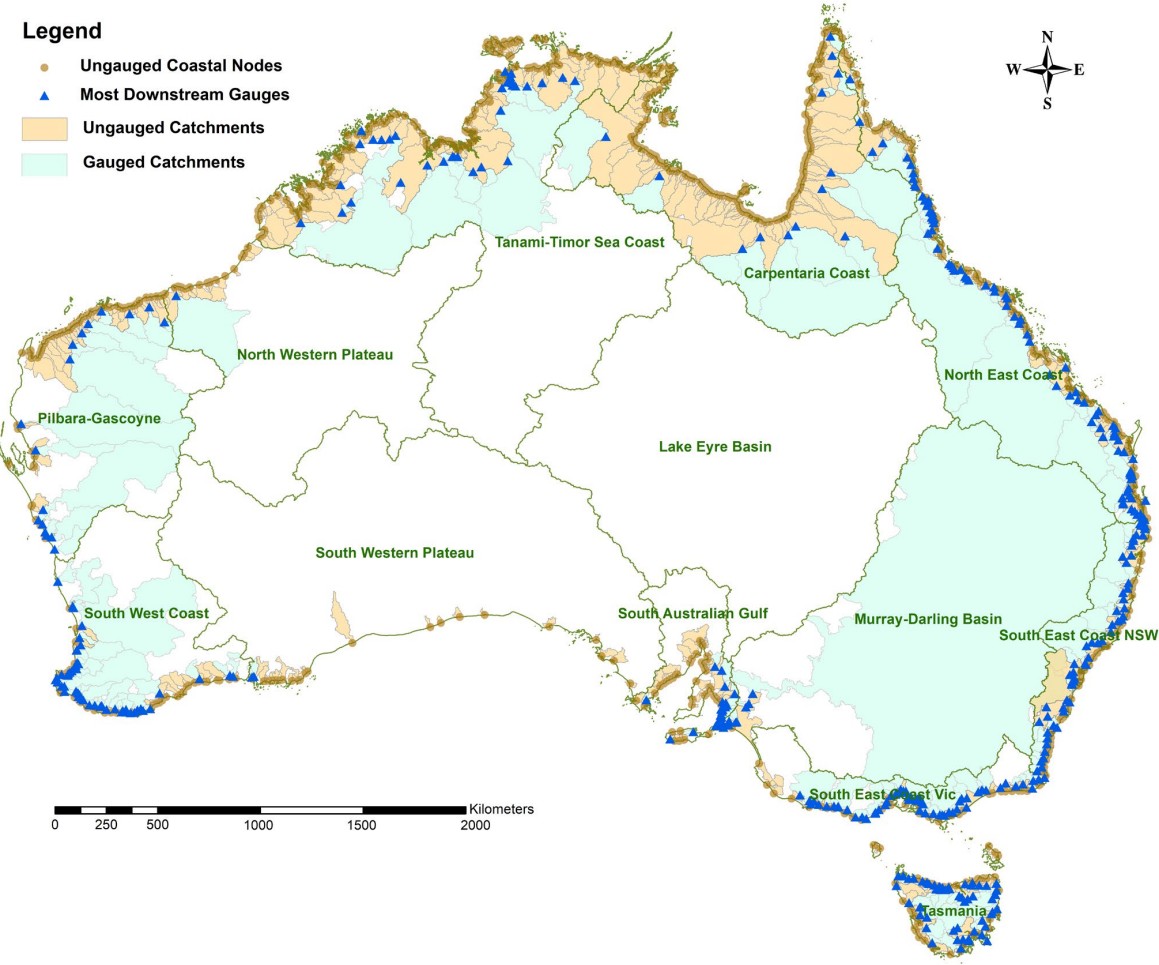

**Figure 1.** Location of gauged and ungauged catchments discharging to the entire Australian coastline.

## 2. Australia's Coastal Regions

More than 80% of Australians live within the coastal regions [7,17]. These includes cities and support important industries such as agriculture, fisheries, and tourism. A significant number of important environmental, biological, and heritage sites are situated within the coastal regions, including wetlands, estuaries, mangroves, and coral reefs. Almost all of the major river systems discharge into the coastal regions, which enrich environmental assets and support the livelihood of most Australians.

### 2.1. Weather and Climate

Droughts, floods, and bushfires are very common in Australia as it is the driest inhabited continent on earth, receiving only 450 mm mean annual rainfall [18]. The rainfall also varies spatially and temporarily across the country, with approximately 70% of the landmass being arid or semi-arid, receiving less than 475 mm per year [19]. Australia's climate zones were defined via Köppen climate classification [20] and have equatorial, tropical, and subtropical regions in the north and temperate regions in the south (Figure A1 in Appendix A). The southeastern and southwestern parts of Australia have temperate climates, and the north has a tropical climate [20]. The eastern and southeastern coastal regions have mountain ranges. Annual rainfall is higher and more reliable in coastal regions, with the exception of the mid-west coastal regions of Western Australia (Figure A2 in Appendix A). Landscape elevation influences the amount and distribution of rainfall,

with mountainous areas such as northeast Queensland, southeast Australia, and western Tasmania receiving higher rainfall [21].

Australia's river system was divided into 13 drainage divisions (Figure 1). Rivers in all drainage divisions discharge into coastal waters and oceans, with the exception of Lake Eyre. The mean annual rainfall of these drainage divisions varies significantly. The mean annual potential evapotranspiration (PET) exceeds the mean annual rainfall except for Tasmania (Figure A2 in Appendix A). The water-limited environment [22] generally controls the streamflow generation processes. The within-year distribution pattern of rainfall, streamflow, and potential evapotranspiration across different drainage divisions in the coastal regions vary widely. The wet season starts in November–December and ends in March–April in the northern part, while in the southern part of the continent, it begins in June–July and ends in December–January, respectively.

### 2.2. Streamflow Measurements

There are approximately 4800 streamflow gauging stations in Australia [23]. These gauging stations are predominantly located in the catchments which have high economic, environmental, social, and cultural significance. At first, we considered all streamflow gauged and ungauged catchments from all 12 drainage divisions draining into the Australian coastline, with the exception of the Lake Eyre drainage division. Our primary focus was on the catchments discharging into the marine environment, which impacts the entire Australian coastline. Most gauging stations are located on the eastern coast of Australia, whereas most of the ungauged catchment area is located on the northern and southern coasts of Australia (Figure A3 in Appendix A). As our primary objective is to create continuous streamflow data at the coastal river nodes (ungauged catchment outlets) located across Australia, we selected gauges based on the following:

- Distance from the coast in the catchment to avoid tidal effects and minimising the ungauged area;
- The availability of data from 1993 onwards with at least 5 years of operational observed streamflow data.

Through this process, we selected a total of 405 most-downstream gauged locations from 12 drainage divisions (Figure 1); other locations which did not meet the above criteria were rejected and categorised as ungauged catchments. The average data length of all 405 stations was 21 years.

### 2.3. Developing Gauged and Ungauged Catchments

We considered gauged and ungauged catchments from all 12 drainage divisions draining into the Australian coastline. First, the Geofabric Australian Hydrological Geospatial Fabric [24] (Bureau of Meteorology website: http://www.bom.gov.au/water/geofabric/, accessed on 24 November 2023) layers were used to delineate all gauged and ungauged catchments. Ungauged catchments were defined as either: (i) the catchments between the most-downstream gauging station(s) and the coastline, mainly the tidal zone, with an area greater than 100 $km^2$; and (ii) catchments along the coastline that do not have streamflow gauging stations directly upstream, with an area greater than 100 $km^2$. The ungauged catchments were delineated for all 12 drainage divisions along the coastal regions. We conceptualised the ungauged catchments into four categories (Figure 2):

- Category 1: Ungauged area was downstream of a gauged catchment;
- Category 2: Ungauged catchments where there were nearby gauged catchments within a radius of up to 50 km;
- Category 3: Ungauged catchments with at least two neighbouring gauged catchments within a 50 km to 250 km radius and in the same Köppen climate zone (Figure A1 in Appendix A);
- Category 4: Ungauged catchments with only one or no neighbouring gauged catchments under a 250 km radius but within the same Köppen climate zone.

All the coastal nodes' gauged and ungauged catchments in all 12 drainage divisions are shown in Figure A3 in Appendix A.

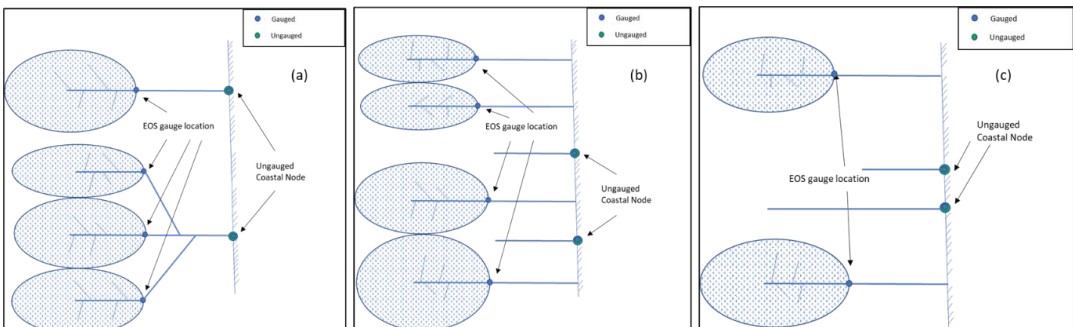

**Figure 2.** Conceptualisation of ungauged catchment: (**a**) Category 1; (**b**) Category 2; and (**c**) Categories 3 and 4.

## 3. Data Quality Control and Gap Filling

### 3.1. Data Quality Control

The observed streamflow data from 405 gauged locations (Figure 1) were sourced from operational Bureau of Meteorology data feeds and directly from the partner organisations. A quality-assurance/quality-control process was required and applied to all datasets used; specifically, observation time-series of daily streamflow. The quality-assurance process involved the identification and removal of erroneous data values such as negative, extreme, and long linear interpolation. The process of detection and removal was automated and then checked manually. An example of an erroneous dataset is shown in Figure 3. The process includes the following steps and was performed manually for the datasets obtained from all the gauged locations:

- Download the time-series dataset and run the QATS (quality assurance of time-series) tool;
- Manually fill missing values (those unobserved and picked up by the tool) through a gap-filling heuristic;
- Plot the time series to manually scan for errors not flagged through automation;
- Reapply the above steps until a final dataset is agreed upon.

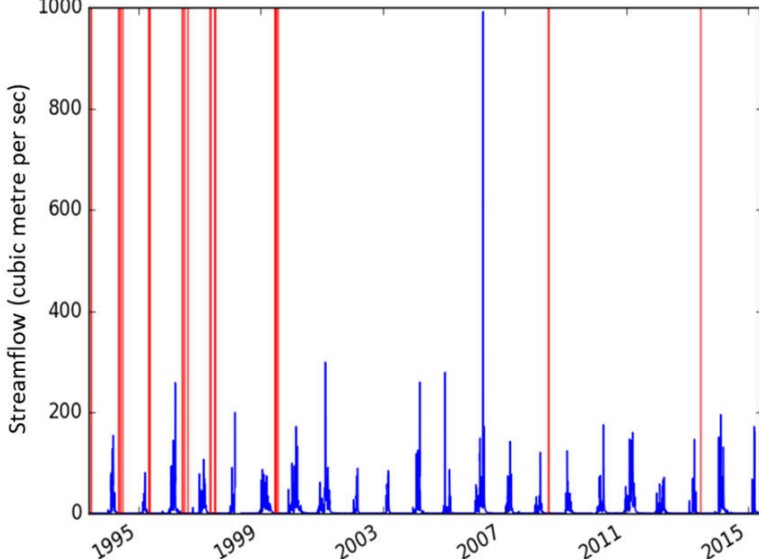

**Figure 3.** An example of the gap-filled streamflow data. The continuous observed streamflow is shown in blue and red lines, indicating poor quality data captured by the QATS (quality assurance of time-series) tool. Depicted is a typical catchment from the southwest of Western Australia.

*3.2. Gap Filling*

The gap filling of the observed discharge datasets was completed through (i) the application of the GR4J model [16]; or (ii) the adoption of the interpolation technique considered most suitable. A gap may constitute missing data, discard erroneous data, or a constant value, or otherwise be picked up by running the automated quality-assurance procedure and by means of manual inspection. If the gap was greater than 5 days, we used the GR4J simulation time series with a simple error correction to fill it. However, if the gap was less than or equal to 5 days, we adopted a three-step procedure in filling these gaps:

- A linear interpolation was applied where the leading or rising trend of the hydrograph appeared to be constant, and little change occurred in the hydrometeorological information of rainfall and/or potential evapotranspiration (PET).
- The GR4J model was applied where a noticeable change appeared in the leading or rising trend of the hydrograph alongside evidence of a variation in the hydrometeorological information of rainfall and/or PET.
- In the case that a linear trend or otherwise was apparent, the gap was checked against the hydrological model simulations for the relevant durations, and where the trend was constant or where no noticeable event was simulated by the model, the linear interpolation technique was adopted or otherwise kept unchanged.

The filling of gaps varied from one catchment to another, i.e., from less than 1% to a maximum of 5% of the data.

## 4. Methodology

In this study, our main objective was to generate the continuous, simulated streamflow time series for gauged and ungauged catchments across the entire coast of Australia using the GR4J hydrologic model, as detailed in Figure 4. We also assessed the performance of the model and identified avenues for future research and development.

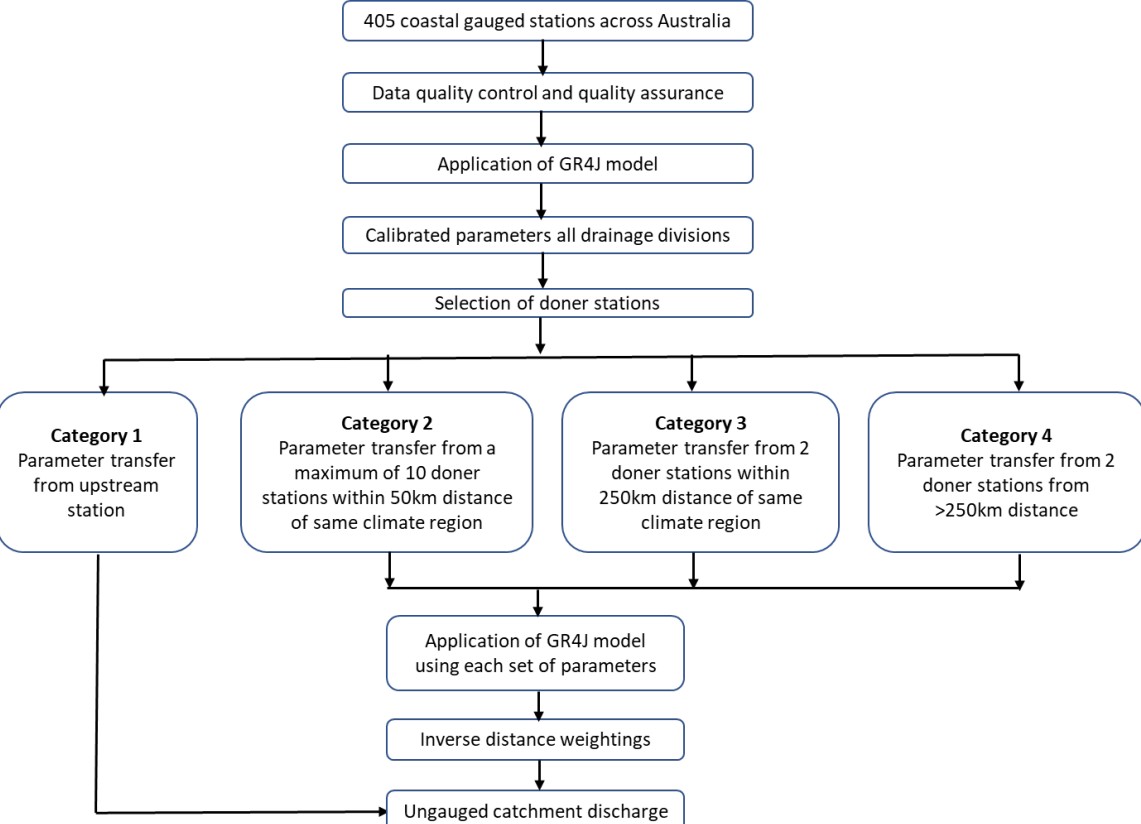

**Figure 4.** Flow diagram for estimating streamflow at ungauged locations.

*4.1. Application of GR4J Model to Gauged Catchments*

The GR4J is a simple four-parameters daily rainfall–runoff model [16]. A schematic of the model is presented in Figure A4 in the Appendix A. It has been included in the Bureau of Meteorology's Short-term Water Information Forecasting Tools (SWIFT) [25]. Research conducted in Australia [16,26–30] has demonstrate that GR4J and its hourly variant (GR4H) perform well in the Australian context. Therefore, we have chosen and applied the GR4J rainfall–runoff model to all gauged catchments. The Australian Hydrological Geospatial Fabric (Geofabric) [24] has a nationally consistent flow direction map. We used Geofabric to delineate each catchment. We conceptualised a catchment, irrespective of its area, spatial distribution of rainfall, and potential evapotranspiration, as one unit, and the fundamental hydrologic model was applied to each catchment; therefore, no routing of the generated streamflow was required. The model was calibrated for each catchment using the Shuffle Complex Evolution—University of Arizona (SCE-UA) algorithm [31].

Some of the 405 catchments have regulated structures including dams, weirs, and water storages. For simplicity, we did not include water balance modelling of water storage inflow, release, spill, draw, diversion, return flow, and evaporation. These infrastructures are operated by different water management entities, and most of the data were not publicly available for model application. As GR4J is a simple model, it has no capacity to represent the effects of these artificial structures on streamflow.

### 4.1.1. Input Data Preparation

The model was developed without a priori knowledge of the rainfall–runoff transformation, with two inputs to the model: (i) areal average daily rainfall; and (ii) daily potential evapotranspiration (PET), both obtained from the Australian Water Availability Project (AWAP) [32]. Daily rainfall and potential evapotranspiration data are available at a 5 km by 5 km grid across Australia. The PET data were estimated using the Priestley–Taylor evaporation equation [33] as part of the AWAP and were available for model application. The average areal observed rainfall and PET for each catchment were calculated by averaging the value of grids (5 km by 5 km) within the catchment. The discharge data for all 405 gauging stations were prepared and quality-checked as detailed in Section 3.1.

### 4.1.2. Objective Function for Model Calibration

The SWIFT modelling suite has inbuilt objective functions that include Nash–Sutcliffe efficiency (NSE) and Kling–Gupta efficiency (KGE) [34]. In this application, we used NSE and KGE separately for GR4J model calibration and presented NSE results for brevity. Additionally, we used diagnostic plots, which provide visual images and empirical understanding of calibrated time series [35,36]. All the model parameters were automatically calibrated using the Shuffle Complex Evolution—University of Arizona (SCE-UA) algorithm [31]. The SCA-UA has been successful in global optimisation based on: (i) a combination of probabilistic and deterministic approaches; (ii) clustering; and (iii) competitive evolution of points in space and direction of global improvement. First three years of observed streamflow data were used for the model 'warm up' period and were not used for calibration optimisation.

*4.2. Estimation of Ungauged Streamflow*

There are plenty of studies on estimating ungauged streamflow. Following decades of research in ungauged basins [6], a few comprehensive reviews of the procedure of estimating ungauged streamflow have been completed: (i) the regionalisation of streamflow, model parameter optimisation, and uncertainty [37]; (ii) rainfall–runoff modelling through identifying hydrological similarity and transposing parameters from gauged to ungauged catchments [38]; and (iii) challenges ahead for cold ungauged regions across the globe [39]. These reviews demonstrate that numerous approaches had already been developed for simulating streamflow time series in ungauged catchments, and rainfall runoff modelling plays a major role [40]. It has been widely used for predicting streamflow times series

in ungauged catchments in Europe [40], the USA. [41,42], Australia [43,44], Canada [39], South America [45], Africa [39,46], and Asia [38,44,47,48].

Various methods have been used in transferring calibrated rainfall–runoff model parameters obtained from gauged to ungauged catchments. There are many studies which have used the entire set of calibrated parameter values from a donor catchment to simulate streamflow of a targeted ungauged catchment. The donor catchment is generally selected based on (i) physical features, similarities, and/or (ii) spatial proximity to the targeted ungauged catchment. It has been demonstrated that the geographically closest catchment (or spatial proximity) to the target ungauged catchment is often the best donor catchment [6,43,49–52]. The parameter regression method has also been used to transfer parameters to ungauged catchments, with the presumption that the calibrated parameters represent catchment attributes (e.g., slope, elevation, drainage density, land use, soil type). In this method, empirical relationships between catchment attributes are obtained and are used to estimate model parameters in ungauged catchments [40,51,53]. Comparison studies show that spatial proximity performs better than the parameter regression method for regions with dense networks of gauging stations [54–56].

In this study, we have used the spatial proximity method in selecting the donor catchments (where the GR4J model was calibrated) to obtain the parameters of the targeted ungauged catchments. At first, we calibrated and applied the GR4J model at all 405 gauged catchments across Australia. To obtain parameters for the ungauged catchments, we conceptualised them into three different categories, as depicted in Figure 4. Then, we estimated model parameters sets and applied the GR4J model to each of the ungauged catchments using each of the parameter sets with estimated sub-areal rainfall and PET, as detailed in the following section. For Category 2–4 ungauged catchments, inverse distance-weightings were applied for final streamflow estimate.

A GR4J parameter transfer method was applied to those catchments in Category 1. The gauged runoff from one or more upstream gauged catchments was routed to the ungauged point and accumulated with the ungauged estimate at a chosen end of the system coastal node (Figure 2a) in the ungauged area. A warmup period of 3 years was applied as part of the modelling procedure. The accumulated output was converted to a discharge time-series, reported at the coastal node.

For the Category 1 ungauged catchment, discharge is estimated by

$$UG_i^1 = \hat{Q}_{us} + \sum_{j=1}^{N} G_j \tag{1}$$

where:

$G_j$ was the gap-filled observed discharge time-series from the gauged locations upstream of an ungauged node on the same river or tributary (Figure 2a);

$\hat{Q}_{us}$ was the simulated discharge from the intermediate area using parameters from the upstream gauge on the same river as the coastal node.

The daily streamflow time-series for ungauged catchments in Category 2 were generated through the parameter transfer of $N$ neighbouring catchments (Figure 2b). $N$ is the number of gauged catchments (up to 10) falling inside a maximum Haversine distance [57] of 50 km from the ungauged catchment in question. The discharges from close-by catchments satisfying the aforementioned conditions were calculated through the application of the GR4J model. The parameters generated from the gauged catchments are used with the PET and rainfall climate data to generate the discharges for the ungauged catchments. Finally, the time-series from this category of catchments was estimated at the coastal node through the inverse-distance weighting of the $N$ time-series. The ungauged area discharge is estimated by

$$UG_i^2 = \sum_{j=1}^{N} \mathbb{1}_{D_j \leq M} \cdot \hat{Q}_j \cdot W_j \tag{2a}$$

$$W_j = \frac{\left(\frac{1}{D_j}\right)^p}{D} \ , \ \ where \ \ D = \sum_{j=1}^{N} \left(\frac{1}{D_j}\right)^p \tag{2b}$$

where:

$M = 50$ and $p = 1$;

$\mathbb{1}_{D_j \leq M = 50}$ was the indicator function, such that if the distance $D_j$ is more than $M = 50$ km, then the time-series is not used to estimate the discharge;

$W_j$ was an inverse distance weighting of power $p$, such that simulated discharge from closer sites receives a larger weighting than those further away.

For each of the ungauged catchments in Category 3, the two nearest gauged catchments where GR4J models were applied for gap filling were selected, such that gauged catchments were within a Haversine distance of 50 km to 250 km of the ungauged area in the same Köppen climate region. The generation of the final estimated time-series at the coastal node was identical to Category 2. Parameters from the two selected gauged catchments and climate data from the ungauged catchment were used to generate two discharge time-series. Finally, a continuous daily discharge for this category of catchments was estimated through inverse-distance-weighting of the two simulated time-series. The ungauged area discharge is estimated by

$$UG_i^3 = \hat{Q}_1 \cdot \left[ \frac{\left(\frac{1}{D_1}\right)^p}{\left(\frac{1}{D_1}\right)^p + \left(\frac{1}{D_2}\right)^p} \right] + \hat{Q}_2 \cdot \left[ \frac{\left(\frac{1}{D_2}\right)^p}{\left(\frac{1}{D_1}\right)^p + \left(\frac{1}{D_2}\right)^p} \right], \tag{3}$$

where:

$M = 250$ and $p = 1$;

$50 \ \text{km} < (D_1, D_2) \leq M = 250$ km.

This was a simplified version of Category 2 with $N = 2$. Where the Haversine distance between the closest two gauged catchments in the same Köppen climate region (Figure A1 in Appendix A) and the ungauged catchment was greater than 250 km, the ungauged catchment was placed in Category 4. The same method outlined in Category 3 was used to estimate discharge from these catchments. The ungauged area discharge is estimated by

$$UG_i^4 = \hat{Q}_1 \cdot \left[ \frac{\left(\frac{1}{D_1}\right)^p}{\left(\frac{1}{D_1}\right)^p + \left(\frac{1}{D_2}\right)^p} \right] + \hat{Q}_2 \cdot \left[ \frac{\left(\frac{1}{D_2}\right)^p}{\left(\frac{1}{D_1}\right)^p + \left(\frac{1}{D_2}\right)^p} \right], \tag{4}$$

where:

$p = 1$;

$50 \langle min(D_1, D_2) \leq 250 \ and \ max(D_1, D_2) \rangle 250$ km or $min(D_1, D_2) > 250$ km.

For each of the categories above, the daily discharge was aggregated to annual and compared between different drainage divisions. It is crucial to mention that the estimated daily discharge data should be used prudently, given the underlying uncertainty of the estimated daily data.

### 4.3. Evaluation Criteria

We have chosen a number of verification metrics and diagnostic plots in evaluating the GR4J model performance, as detailed in the following sections.

#### 4.3.1. Evaluation Metrics

There are many goodness-of-fit criteria for hydrological model calibration and performance assessment [58]. For the performance evaluation of the GR4J model at all observed streamflow locations, we used Nash–Sutcliffe efficiency [59] and the percent bias (PBias). as presented in Table 1. We also used the coefficient of determination ($R^2$) between the

calibrated and observed streamflow time series for gauge locations selected for diagnostic plots. Moriasi et al. [60] and Chiew and McMahon [61] recommended that model performance is considered good when the *NSE* is greater than 0.5 and the PBias ranges are less than ±25% for monthly streamflow. However, *NSE* values lower than 0.5 for daily streamflow can still be considered satisfactory. Therefore, some of the constraints for the recommended statistics [61] can be relaxed for daily streamflow, but in our study, we kept the performance identical to the monthly streamflow. Keeping the constraints identical to monthly flow resulted in greater confidence in daily model performance.

**Table 1.** Metrics used for model performance evaluation.

| Metrics | Abbreviation | Equation | Description |
|---|---|---|---|
| Nash-Sutcliffe Efficiency | NSE | $NSE = 1 - \dfrac{\sum_{i=1}^{n}\left(Q_{i,obs}-Q_{i,sim}\right)^2}{\sum_{i=1}^{n}\left(Q_{i,sim}-\overline{Q_{obs}}\right)^2}$ | Compares the mean square error against the observation variable. It varies between $-\infty$ to 1 with a perfect score of 1. |
| Percent bias | PBias | $PBias = \dfrac{\sum_{i=1}^{n}\left(Q_{i,obs}-Q_{i,sim}\right)}{\sum_{i=1}^{n}Q_{i,obs}} * 100$ | Measures the difference between the mean/median of forecast variable and observation. It varies between $-\infty$ to $+\infty$ with a perfect score of 0. |

### 4.3.2. Evaluation Diagnostic Plots

Diagnostic plots generally provide visual images of the model performance metrics and also provide empirical understandings of model calibrated time series [35,36]. We have chosen three popular diagnostic plots, i.e., times series, flow-duration, and correlation scatter plots, for the evaluation of model performance (Table 2).

**Table 2.** Diagnostic plots used for model performance evaluation.

| Plot | X-Axis | Y-Axis | Description |
|---|---|---|---|
| Time series | Time step | Simulated and observed streamflow | Daily and monthly discharge |
| Flow-duration | Probability of exceedance (%) | Simulated and observed streamflow | Daily and monthly streamflow |
| Correlation scatter | Observed streamflow | Simulated streamflow | Daily, monthly, and annual total streamflow |

### 4.3.3. Model Performance Ratings

In this study, we used the model evaluation metrics NSE and PBias statistics for the daily streamflow for the gauged catchments. These metrics were used by Kalin et al. [62], Yilmaz and Onoz [47], and Chen et al. [63]. Based on these two metrics, model performance on daily streamflow is characterised as 'Very good', 'Good', 'Satisfactory', and 'Unsatisfactory' (Table 3).

**Table 3.** Performance ratings of Nash–Sutcliffe efficiency (NSE) and percent bias (PBias) statistics for daily streamflow.

| Performance Rating | NSE | Catchment (%) | Abs (PBias) % | Catchment (%) |
|---|---|---|---|---|
| Very Good | NSE ≥ 0.70 | 57 | Abs(PBias) ≤ 25 | 88 |
| Good | 0.5 ≤ NSE < 0.7 | 23 | 25 < Abs(PBias) ≤ 50 | 6 |
| Satisfactory | 0.3 ≤ NSE < 0.5 | 8 | 50 < Abs(PBias) ≤ 70% | 3 |
| Unsatisfactory | NSE < 0.3 | 12 | Abs(PBias) > 70% | 3 |

## 5. Results

### 5.1. Gauged and Ungauged Catchments

The total gauged catchment area comprises 405 stations across the coastal regions of Australia and has an area of 2,549,000 km$^2$ (Table 4). A number of catchments where gauged streamflow data presented water balancing issues, mainly due to return flows and diversions, were excluded from the estimation. There was a total of 771 ungauged catchments, categorised as follows: Category 1 (183 catchments); Category 2 (212 catchments);

Category 3 (228 catchments); and Category 4 (148 catchments). The number of ungauged catchments and their areas of different categories varied from one drainage division to another (Table 4). The CC drainage division has the largest ungauged catchment area. The total ungauged catchments have an area of 835,000 km$^2$ and represent 24.7% of the total drainage division areas. The TTS and CC drainage divisions have the largest ungauged area. Maps of gauged and ungauged catchments in each of the drainage divisions are shown in Figure 1 and detailed in Figure A3 in the Appendix A.

**Table 4.** List of gauged and ungauged catchments and areas (1000 km$^2$) in each of the drainage divisions.

| Drainage Division | Gauged Stations | | Ungauged Area | | | | |
| --- | --- | --- | --- | --- | --- | --- | --- |
| | No. | Area | 1 | 2 | 3 | 4 | Total |
| Northeast Coast (NEC) | 83 | 366 | 35 | 13 | 10 | 0 | 58 |
| Southeast Coast NSW (SEN) | 60 | 75 | 44 | 4 | 2 | 0 | 50 |
| Southeast Coast VIC (SEV) | 60 | 75 | 11 | 4 | 1 | 0 | 16 |
| Tasmania (TAS) | 53 | 38 | 21 | 1 | 0 | 0 | 22 |
| Murray–Darling Basin (MDB) | 7 | 882 | 9 | 0 | 0 | 0 | 9 |
| South Australian Gulf (SAG) | 23 | 9 | 5 | 5 | 8 | 6 | 24 |
| Southwest Coast (SWC) | 55 | 159 | 8 | 8 | 4 | 0 | 21 |
| Pilbara–Gascoyne (PG) | 17 | 276 | 11 | 19 | 18 | 3 | 52 |
| Tanami–Timor Sea Coast (TTS) | 33 | 312 | 91 | 21 | 95 | 26 | 233 |
| Carpentaria Coast (CC) | 13 | 304 | 141 | 21 | 81 | 73 | 315 |
| Northwestern Plateau (NWP) | 1 | 53 | 0 | 6 | 3 | 7 | 17 |
| Southwestern Plateau (SWP) | 0 | 0 | 2 | 1 | 4 | 11 | 18 |
| Total | 405 | 2549 | 378 | 106 | 231 | 128 | 835 |

*5.2. Model Calibration and Validation*

We calibrated and validated the daily G4R4J model to all gauged catchments for the period 1993 onwards. The daily discharge from one drainage division to another varies significantly due to catchment landscape attributes, within year distribution of rainfall, and PET [23,30]. The calibrated model parameters varied from one catchment to another but were within the range of previous application in Australia [30] and around the world [64,65]. We present observed and simulated daily streamflow hydrographs and flow–duration curves of three catchments, located in the TAS, SWC, and SAG drainage divisions (Figure 5). In some instances, simulated high flows were earlier or later compared to the observed streamflow. These catchments present a balanced view of the model performance as defined in Table 2. The simulated daily streamflow, high and medium range, generally matched well with the observed streamflow. However, the low flow was generally overpredicted, as is evident in the flow–duration curves (Figure 5). This may be explained by (i) oversimplification of process representation by conceptualising it as one system irrespective of the catchment area; (ii) inability to represent spatial variability of rainfall, PET, and catchment attributes; and (iii) absence of channel routing.

We also present the scatter plots of simulated and observed daily streamflow—one from each of the drainage divisions (Figure 6). As with the daily hydrographs (Figure 5), these catchments present a balanced view of the model performance. In some cases, the simulated high flows were lower than the observed, or the timing was earlier or later, which resulted in 'Unsatisfactory' NSE (Table 3).

The model calibration results, NSE and PBias, for all gauged catchments in each of the drainage divisions are presented in Figure 7. The model calibration was rated as 'Very good' for 57% and 88% of the catchments based on NSE and PBias, respectively (Table 3). However, the range of these two metrics varied significantly for different catchments within and between the drainage divisions (Figure 7). The MDB, SAG, and PG drainage divisions had the highest range of NSE—from 0.05 to 0.95—but the PBias was lower. The NEC drainage division had a higher range of NSE and PBias distribution (Figure 7).

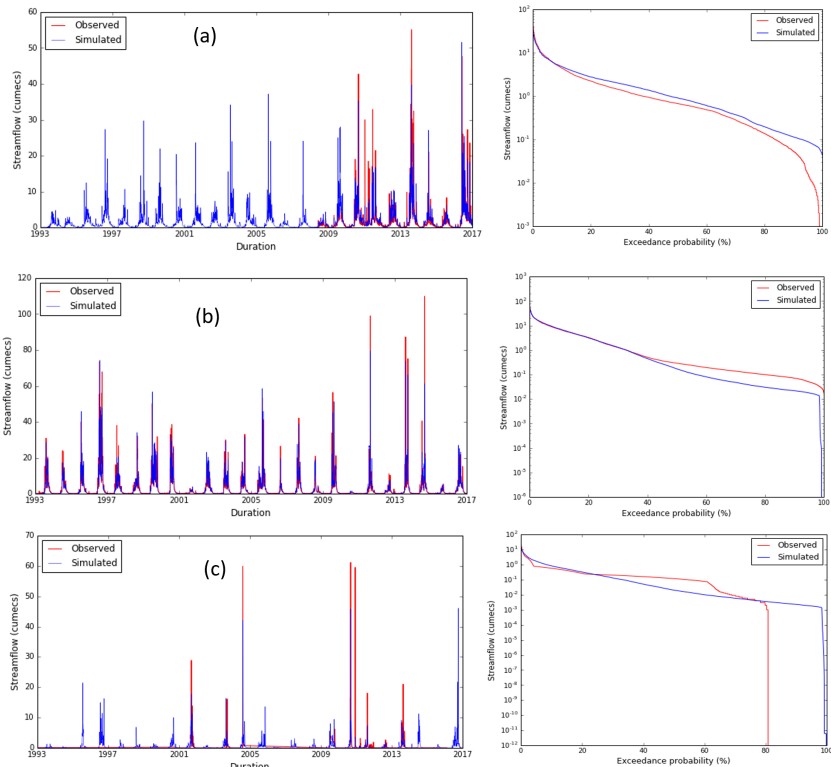

**Figure 5.** Daily observed and simulated streamflow hydrographs and flow–duration curves—typical catchments from (**a**) TAS, (**b**) SWC, and (**c**) SAG drainage divisions.

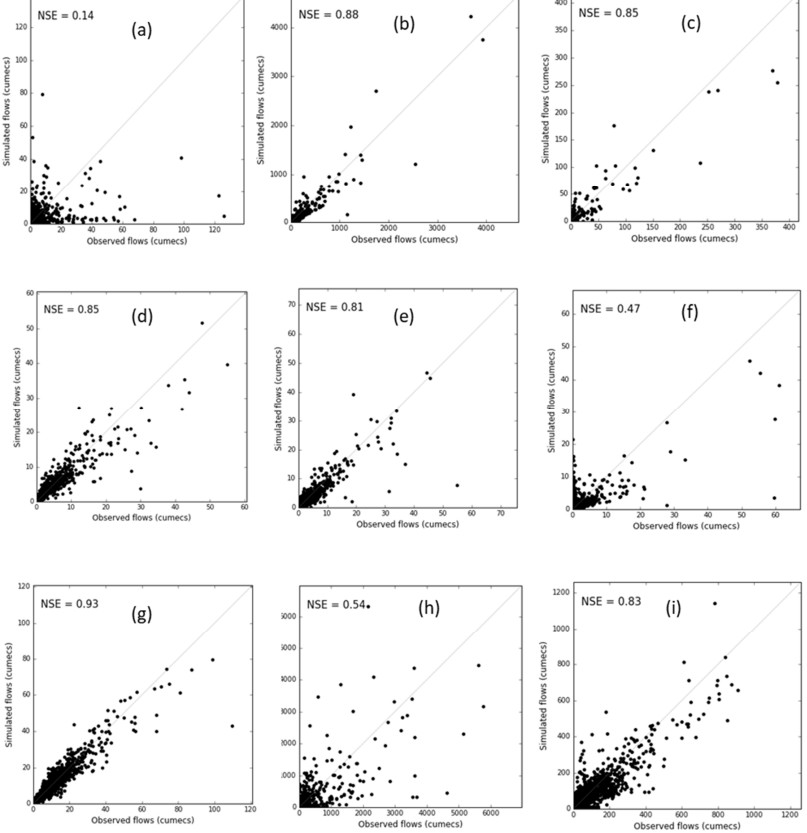

**Figure 6.** Daily streamflow scatter plots—a typical catchment from: (**a**) NEC; (**b**) SEN; (**c**) SEV; (**d**) TAS; (**e**) MDB; (**f**) SAG; (**g**) SWC; (**h**) PG; and (**i**) CC drainage divisions.

We also investigated the model calibration and performance metrics, NSE and PBias, and their relationship with catchment physical attributes in particular catchment areas. Our results show that no strong relationship exists between catchment areas and both metrics, i.e., NSE and PBias (Figure 8).

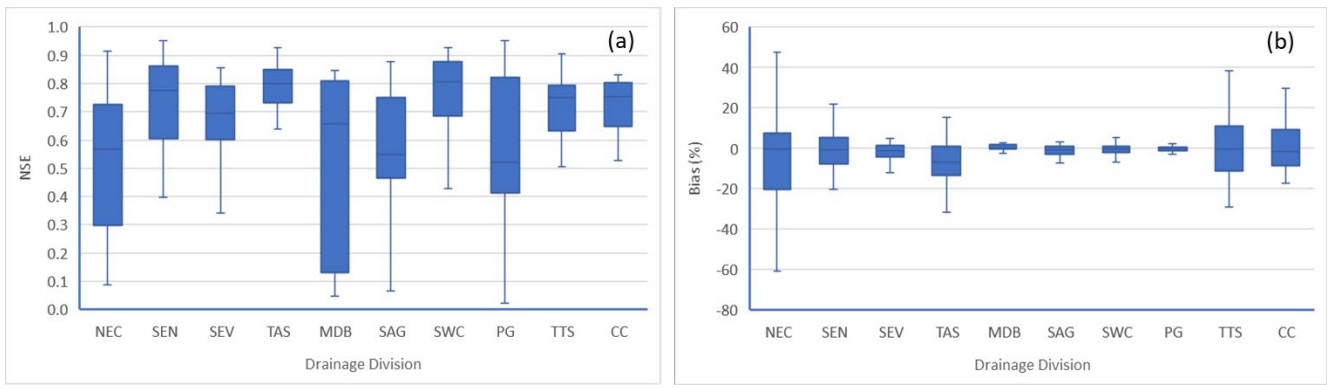

**Figure 7.** Calibrated (**a**) NSE and (**b**) PBias of catchments within all drainage divisions.

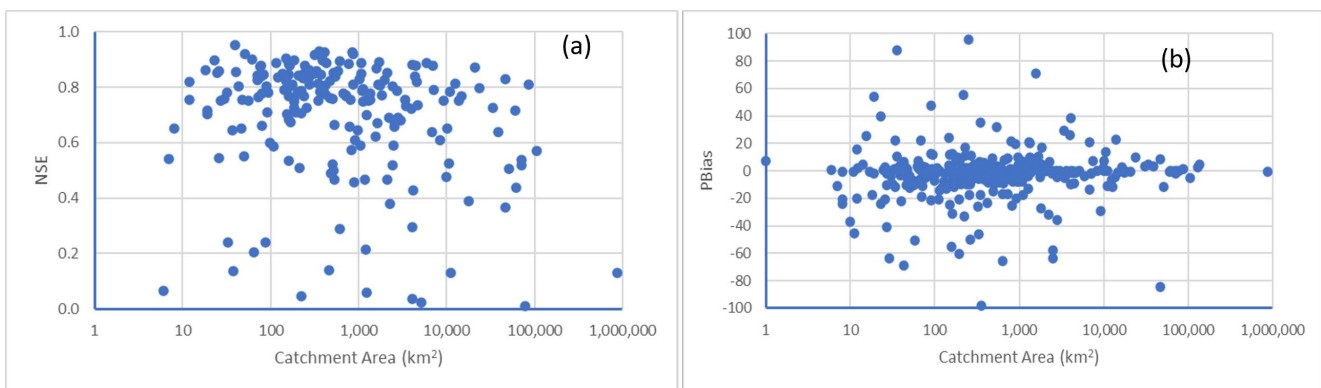

**Figure 8.** Relationship between (**a**) NSE and (**b**) PBias for all gauged catchment areas.

### 5.3. Performance Evaluation—Gauged Catchments

We evaluated the performance of the GR4J model based on the evaluation criteria presented in Section 4.3.1. Visual inspection of diagnostic plots, including daily hydrographs, flow–duration curves, and scatter plots were completed in evaluating each of the models. A general visual agreement between the observed and simulated streamflow indicates adequate calibration and validation, which represent catchment processes and the model's ability to reproduce hydrological behaviors [58]. Most of the models represented the catchment process well, but some were 'Unsatisfactory', as evident through NSE and PBias metrics. Based on NSE only, 57% of the model performance was rated 'Very good', 23% 'Good', 8% 'Satisfactory', and 12% 'Unsatisfactory', respectively. However, according to the PBias metric, 88% of the models were rated as 'Very good', 6% 'Good', and only 3% 'Unsatisfactory' (Table 4). For some catchments, conflicting performance ratings were found—one may be rated 'Very good' or 'Satisfactory' based on the NSE and PBias criteria, respectively. At the drainage division scale, as is evident in Figure 7, for example, the NSE of different catchments within the PG drainage divisions ranged from 0.03 to 0.95, while PBias was only ±5%.

### 5.4. Performance Evaluation—Ungauged Catchments

We assessed the performance of the model in simulating streamflow from ungauged catchments by comparing runoff ratios as the catchment area, flow generation process, PET, and rainfall vary significantly across Australia. For a Category 1 ungauged catchment, there was only a gauged catchment upstream (Figure 2a). The proportion of the gauged

and ungauged catchment areas varied from one catchment to another due to the unique stream network of the ungauged areas. The proportion of ungauged catchment areas ranged from 1% to 95% among all gauged catchments, respectively (Figure 9). However, estimated proportional discharge from ungauged areas was not always similar to the proportion of gauged catchment discharge, mainly due to higher rainfall and lower PET (Figure A2 in Appendix A) in the coastal regions compared to inland gauged areas. This feature was also evident in the runoff coefficients (proportion of runoff and amount of rainfall of a catchment) of Category 1 catchments across all drainage divisions. Compared to the gauged catchments, estimated runoff coefficient distributions from all categories of ungauged catchments within a drainage division were generally greater (Figure 10). This feature was evident across most of the drainage divisions, with the exception of SEV and SWC.

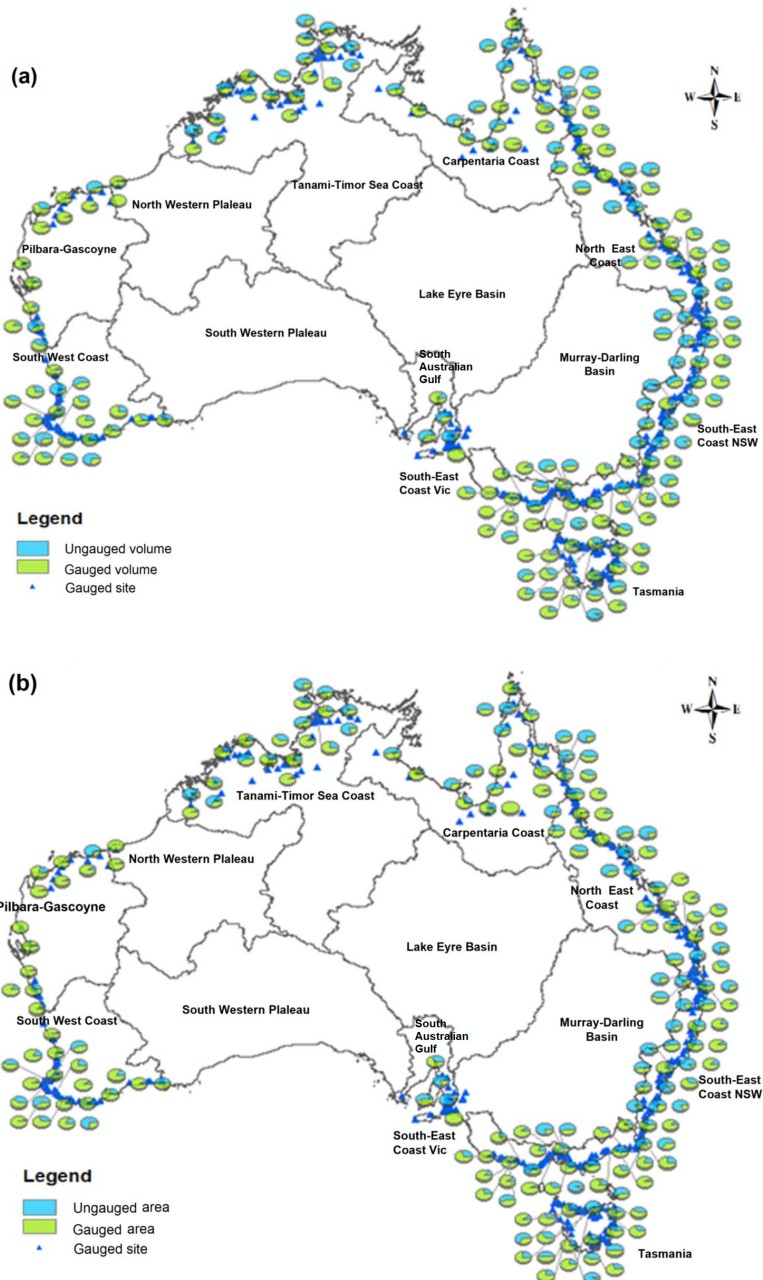

**Figure 9.** Proportion of gauged and ungauged (**a**) streamflow volumes and (**b**) catchment areas—Category 1 catchments.

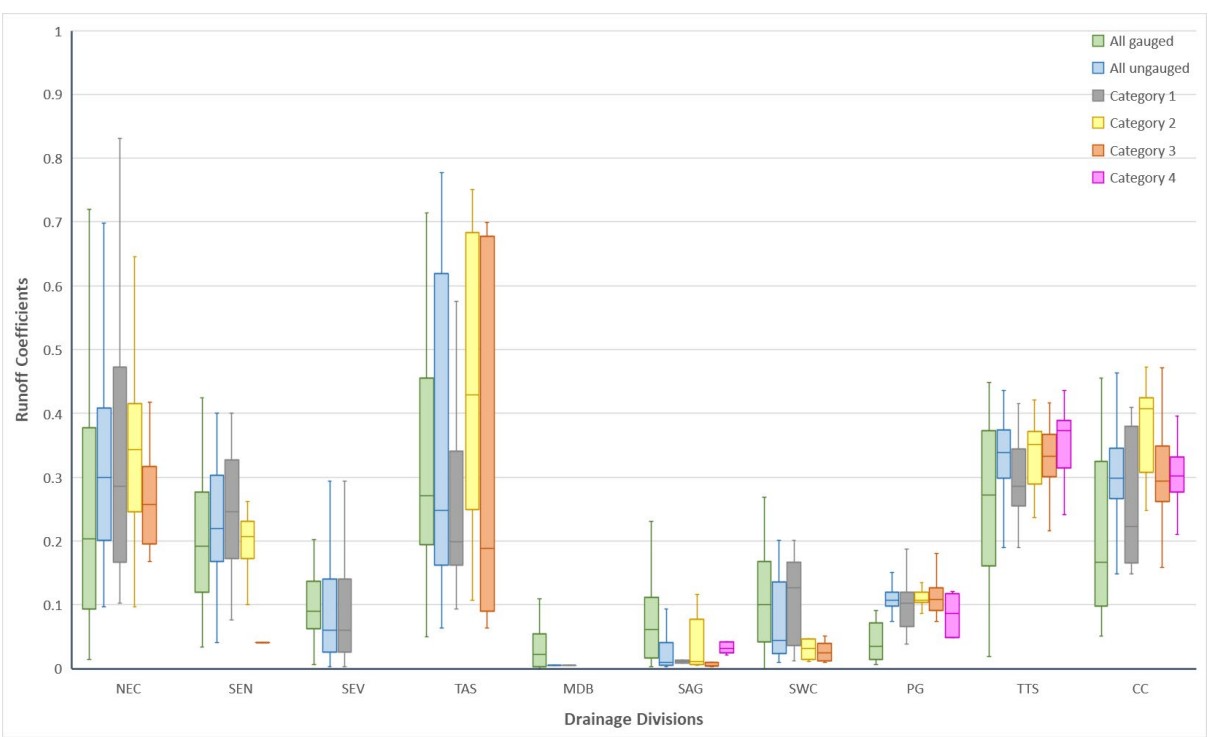

**Figure 10.** Observed and simulated mean annual runoff ratio for all ungauged and gauged catchments in all drainage divisions.

*5.5. Estimated Coastal Discharge*

We estimated the mean annual discharge to Australian coastal regions through the application of the GR4J daily model to the gauged and ungauged catchments. The estimated mean annual streamflow varied significantly from one drainage division to another (Table 5). The mean annual discharge from all drainage divisions, including the gauged and ungauged catchments, was 419,950 gigalitres (GL), with an ungagged catchment contribution of 232,200 GL, representing 55% of the total. Our findings compare well with the National Land and Water Resources Audit [66] estimate of 387,184 GL. The Murray–Darling Basin, Australia's food bowl, has only about 1% catchment area ungauged (Table 3). It is also a very highly managed system, represents 55% of Australia's water use (http://www.bom.gov.au/water/nwa/2020/mdb/regiondescription/geographicinformation.shtml, accessed on 12 November 2023), and only a small proportion of the streamflow reaches the ocean. Estimated discharge from the ungauged catchments was not significant (Table 5). The mean annual streamflow from the NEC drainage division was estimated at 58,470 GL, with an ungauged area contribution of 22,800 GL. This finding compares well with the CSIRO [67] runoff estimates. Estimated streamflow from ungauged areas of SAG and SWC were not significant compared to that of gauged areas. There are no gauging stations, and very limited rainfall gauges exist within SWP. In this drainage division, stream networks are also not well-formed; therefore, estimates of streamflow are very preliminary. The estimated average annual streamflow from gauged and ungauged areas of SWC was 3480 GL (Table 5), which compares fairly with gauged streamflow estimates of 4700 GL [68]. The difference could be due to different periods of data and recent below-average rainfall in Western Australia [69]. The ungauged areas in the PG and NWP were 16% and 24%, respectively. However, there is only one streamflow gauging station in NWP (Figure 1). Estimates of mean annual runoff from gauged catchments within these two drainage divisions range from 2% to 9% (Figure 10). Thius compares well with the Pilbara water resources assessment study covering part of these two drainage divisions [70]. The TTS and CC drainage divisions have the largest proportion of ungauged areas—43% and 51%, respectively (Figure 1, Table 3). Mean annual runoff from the TTS drainage division spatially

varies significantly from 2% to approximately 45% (Figure 10), which compares well with CSIRO [71] finding of 3–40% of all gauged catchments. In the CC drainage division, mean annual runoff was slightly lower than that of TTS and ranged between 3–60%. However, estimates of runoff from ungauged areas seemed to be high compared to their gauged counterparts, ranging from 15 to 45% (Figure 10), probably due to an oversimplified application of the GR4J model. The mean annual streamflow from CC drainage division of 109,440 GL compared well with the CSIRO [71] of 90,000 GL.

**Table 5.** Mean annual discharge (1000 GL) from each of the drainage divisions including gauged and ungauged catchments.

| Drainage Division | Overall Total | Gauged | Ungauged | | | | |
| --- | --- | --- | --- | --- | --- | --- | --- |
| | | | 1 | 2 | 3 | 4 | Total |
| Northeast Coast (NEC) | 58.47 | 35.67 | 11.73 | 7.34 | 3.73 | | 22.80 |
| Southeast Coast NSW (SEN) | 20.4 | 11.23 | 8.04 | 1.07 | 0.06 | | 9.18 |
| Southeast Coast VIC (SEV) | 11.95 | 10.87 | 0.92 | 0.11 | 0.05 | | 1.07 |
| Tasmania (TAS) | 39.07 | 25.12 | 13.10 | 0.85 | | | 13.95 |
| Murray–Darling Basin (MDB) | 4.38 | 4.38 | 0.00 | | | | 0.00 |
| South Australian Gulf (SAG) | 0.23 | 0.04 | 0.02 | 0.11 | 0.02 | 0.04 | 0.19 |
| Southwest Coast (SWC) | 3.48 | 2.57 | 0.73 | 0.15 | 0.03 | | 0.91 |
| Pilbara–Gascoyne (PG) | 6.15 | 4.52 | 0.34 | 0.70 | 0.50 | 0.09 | 1.63 |
| Tanami–Timor Sea Coast (TTS) | 146.15 | 61.72 | 27.16 | 9.69 | 38.12 | 9.46 | 84.43 |
| Carpentaria Coast (CC) | 109.44 | 30.25 | 25.00 | 8.99 | 26.52 | 18.68 | 79.19 |
| Northwestern Plateau (NWP) | 9.41 | 1.48 | 7.38 | 0.03 | 0.28 | 0.24 | 7.93 |
| Southwestern Plateau (SWP) | 10.91 | 0.00 | 10.83 | 0.01 | 0.02 | 0.05 | 10.90 |
| Total | 419.95 | 187.85 | 105.3 | 29.0 | 69.3 | 28.5 | 232.2 |

## 6. Discussion and Future Research

### 6.1. Model Calibration and Performance

We generated the continuous daily streamflow time series for gauged and ungauged catchments across the entire Australia from 1993 onwards through the application of GR4J model. For simplicity, we conceptualised each catchment as one unit and did not subdivide it into sub-catchments and sub-areas to represent the spatial distribution of rainfall, PET, and catchment physical attributes. The performance of the models was analysed based on the performance metrics (Table 1) and visual inspection of daily hydrographs, flow duration curves, and scatter plots. Based on the NSE and PBias performance scores, 12% and 3% of the models, respectively, were categorised as unsatisfactory (Table 3). One explanation could be the model's inability to represent spatial variability of rainfall, evaporation, catchment attributes, and channel routing, and another cause could be the strong influence of high flows on NSE values. Recent application of the GR4J model over 100 catchments across Australia [30,72] demonstrates better performance, including high and low flows, when spatial variabilities and proper channel routing were adopted. Similar results were also found by Viney et al. [73], Zhang and Chiew [43], and Oudin et al. [74] for estimating streamflow from ungauged catchments and selecting donor catchments. Further research may reveal the fundamental causes of these conflicting performance ratings.

We also did not find any relationship between the catchment area and model performance score. Similar results were also found by Silberstein et al. [75] when applying a set of lumped catchment models in southwestern Western Australia. However, Sleziak et al. [76] found a positive correlation between increasing NSE and catchment area when assessing the effectiveness of calibrating conceptual hydrological model in relation to catchment characteristics in Austria. Further research and investigations may reveal the definitive relationship between model performance metrics and catchment areas, particularly in Australia.

### 6.2. Discharge Estimates from Ungauged Catchments

The mean annual discharge from all drainage divisions was 419,950 GL, with an ungagged catchment contribution of 232,200 GL. Our estimates compare well with the National Land and Water Resources Audit [66] finding of 387,184 GL. However, estimates of streamflow from ungauged catchments vary from one drainage division to another (Figure 10, Table 5). But in general, this compare well with various studies accomplished by CSIRO [67,68,70,71].

In this study, we used a maximum of 10 donors for the Category 2 ungauged catchments. But for the other two categories, only two nearest neighbours were considered. Streamflow averaging from multiple-donor catchments consistently gives better estimates of ungauged streamflow than the use of single-donor catchments. However, enhancement of ungauged streamflow estimates generally diminishes as the number of donor catchments increases [77]. It was found that in Australia, up to five donor catchments significantly increase ungauged catchment streamflow estimates [44]. In a study using different types of catchments across the world, it was found that the use of up to 10 similar donor catchments enhanced simulated discharge at the ungauged catchments; even substantial improvements were evident if the donor catchments were from similar climate zones greater than 5000 km away [78]. In a comprehensive study using 671 catchments with diverse hydroclimatology, it was found that a 'perfect' donor catchment exists, but that it is not necessarily the nearest neighbour [51].

The catchment physical similarity approach is another well-known technique used in estimating ungauged streamflow. The application of this approach in Australian catchments may give better outcomes than the nearest neighbour approach [43]. Similar results were also found in Europe and the USA [51,54]. An in-depth novel similarity approach was used by Narbondo, et al. [45], where relationships between GR4J parameters and catchment physical attributes were found and then exported to ungauged catchments to estimate streamflow. This approach consistently provided very satisfactory results and could be adopted for estimating ungauged streamflow with highly variable hydroclimatology. In this study, our scope was limited to using nearest neighbour catchments in estimating GR4J model parameters and thereby estimating ungauged streamflow. In the future, other approaches should be explored [51], including linking GR4J model parameters with catchment physical properties.

There are several other sources of errors that may shape the estimation of streamflow from ungauged catchments. These include errors in the observed datasets, model structural errors and uncertainty in the regionalisation of model parameter sets. Despite these limitations, hydrological modelling is regarded as the most reliable approach to estimating streamflow from ungauged catchments [6,40].

### 6.3. Future Research

In this study, our scope was limited to conceptualising a catchment, irrespective of its area, as one unit, without dividing it into smaller subareas, to represent spatial variabilities including rainfall, PET, and catchment physical attributes. Operational application of the GR4J model in high-value water resource catchments across Australia [30,72] demonstrates that better model calibration performance could be achieved through the spatial representation of catchment variabilities and the adopting of proper channel routing of the streamflow volume generated. Further development could include reservoir water balance, diversion, and return flow. Due to this simplified conceptualisation of a catchment, we did not test the model performance of estimating ungauged discharge, assuming gauged donor catchments as ungauged for each of the categories. Recent research shows that relationships between GR4J parameters and catchment physical attributes exist and could be exported to better estimate ungauged streamflow [45]. In future, this approach should be explored further, including other novel ideas proposed by Pool et al. [51].

## 7. Summary and Conclusions

There are 405 gauged catchments in the coastal regions across Australia that cover 2,549,000 km² across all 12 drainage divisions, and 771 ungauged catchments that cover an additional area of 835,000 km². The distribution of ungauged catchments varies from one drainage division to another, with the largest proportion of 51% in the Carpentaria Coast (CC). The total area draining to the Australian coastal region is estimated at 3,384,000 km². The annual rainfall and PET and their spatial and temporal distribution vary significantly from one drainage division to another.

We generated the continuous daily streamflow time series for gauged and ungauged catchments across all of Australia from 1993 onwards. We applied the GR4J model and assessed its performance using NSE and PBias metrics. Based on these two metrics, the performance ratings of 80% and 96% of the models were classified as good, and only 12% and 3% of the models were unsatisfactory, respectively. We found no relationship between catchment area and model performance, particularly with respect to NSE and PBias.

We categorised ungauged coastal catchments into four categories based on distance and Köppen climate zone: (i) downstream of a gauged catchment; (ii) gauged catchments within a radius up to 50 km; (iii) at least two gauged catchments within a 50 km to 250 km radius and in the same Köppen climate zone; and (iv) one or no neighbouring gauged catchments beyond a 250 km radius but within the same climate zone. The total ungauged catchments have an area of 835,000 km² and represent 24.7% of the total drainage division areas.

We estimated streamflow for ungauged catchments based on the parameters of their donor catchments. Overall, runoff ratios from ungauged catchments were generally higher compared to their donor-gauged catchments due to higher rainfall and less PET in the coastal areas. In particular, this tendency was evident in the CC and Tanami–Timor Sea Coast (TTS) drainage divisions, where ungauged areas comprised 51% and 43%, respectively.

We estimated the mean annual streamflow from each of the drainage divisions based on the application of the GR4J model and its extension to ungauged catchments. The mean annual gauged streamflow varied significantly across different drainage divisions—from 230 GL in the South Australian Gulf (SAG) to 109,440 GL in CC. The estimated mean annual streamflow from all ungauged catchments was 232,170 GL, slightly higher than other estimates, likely due to different methodologies used, including the simplified application of the GR4J model. Overall, the mean annual streamflow from all drainage divisions, including gauged and ungauged areas, across the coastal regions of Australia was estimated at 419,950 GL and compared well with the National Land and Water Resources Audit estimate. The comprehensive streamflow estimates will be helpful in furthering our understanding of coastal processes, models, and tools.

**Author Contributions:** M.A.B. contributed to conceptualisation, investigation, methodology, project administration, resources, supervision, validation, and writing. U.K. undertook data analyses, investigation, and visualisation. G.E.A. undertook data curation, analyses, investigation, validation, and visualisation. R.M.L. contributed to analyses, review, and editing. M.T. undertook data curation, analyses, validation, review, and editing. All authors have read and agreed to the published version of the manuscript.

**Funding:** Some of the initial streamflow data were developed under an agreement with University of Tasmania. However, the University of Tasmania had no role in (i) the design of research; (ii) collection, analyses, and interpretation of data; (iii) writing the manuscript; and (iv) the decision to publish the results.

**Data Availability Statement:** The codes, scripts, and data used for this research are not available to the public. The data are not publicly available due to Bureau's operational policy and procedure.

**Acknowledgments:** We acknowledge the Water Information Research and Development Alliance (WIRADA) for the SWIFT model development and research. We would like to express our sincere thanks to our technical reviewers, Mohammad Hasan, Fitsum Woldemeskel, Alex Cornish, and

Hamideh Kazemi for their review, valuable comments, and suggestions. The technical analysis, advice, and management support received from Narendra Tuteja and Daehyok Shin are sincerely acknowledged. We acknowledge the catchment delineation work conducted by Nilantha Gamage. The computations in this study were conducted using the facilities provided by the National Computational Infrastructure (NCI) supported by the Australian Government. We acknowledge the support and advice from the University of Tasmania, as some of the streamflow data were developed under an agreement and were provided to the marine virtual library (MARVL). We sincerely thank the three anonymous reviewers for their careful review, positive comments, and suggestions. Their contribution definitely enriched the manuscript.

**Conflicts of Interest:** The authors declare that the research was conducted in the absence of any commercial or financial relationships that could be construed as a potential conflict of interest.

**Appendix A**

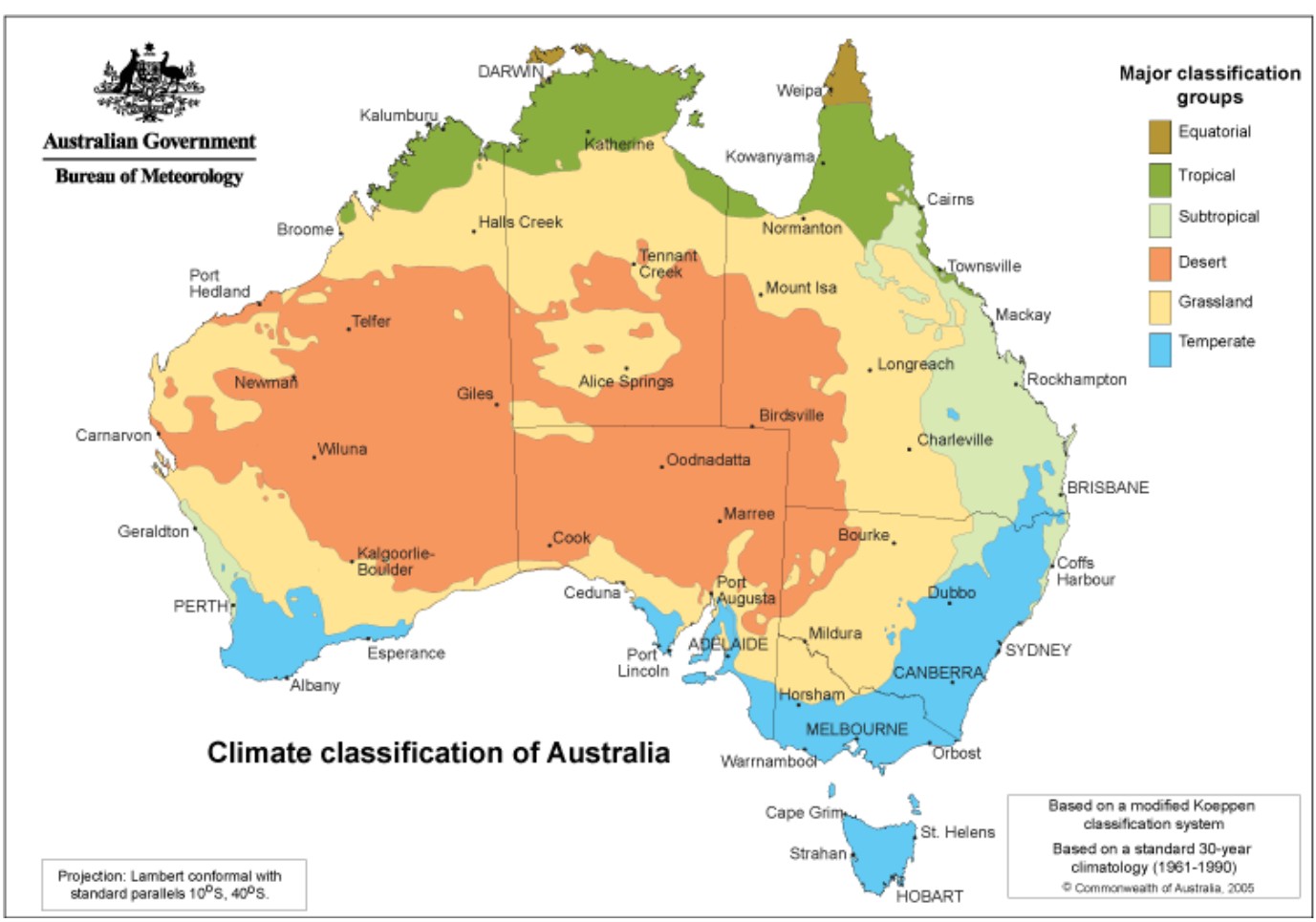

**Figure A1.** The Köppen classification map showing six major groups of climate zones across Australia. These climate zones are defined with the climatic limits of native vegetation in mind. This method of classification is based on the concept that native vegetation is the best expression of climate in an area. The six major classes are identified predominantly on native vegetation type (Bureau of Meteorology: http://www.bom.gov.au/climate/maps/averages/climate-classification/, accessed on 24 May 2023).

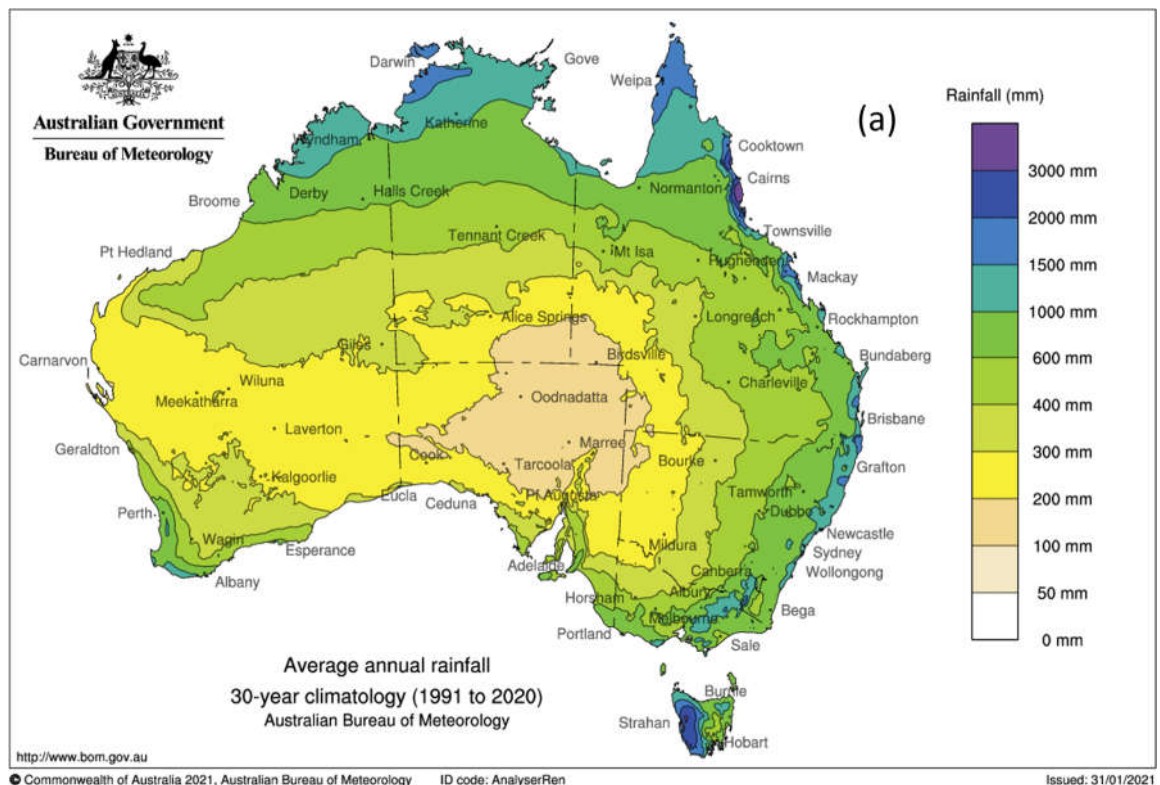

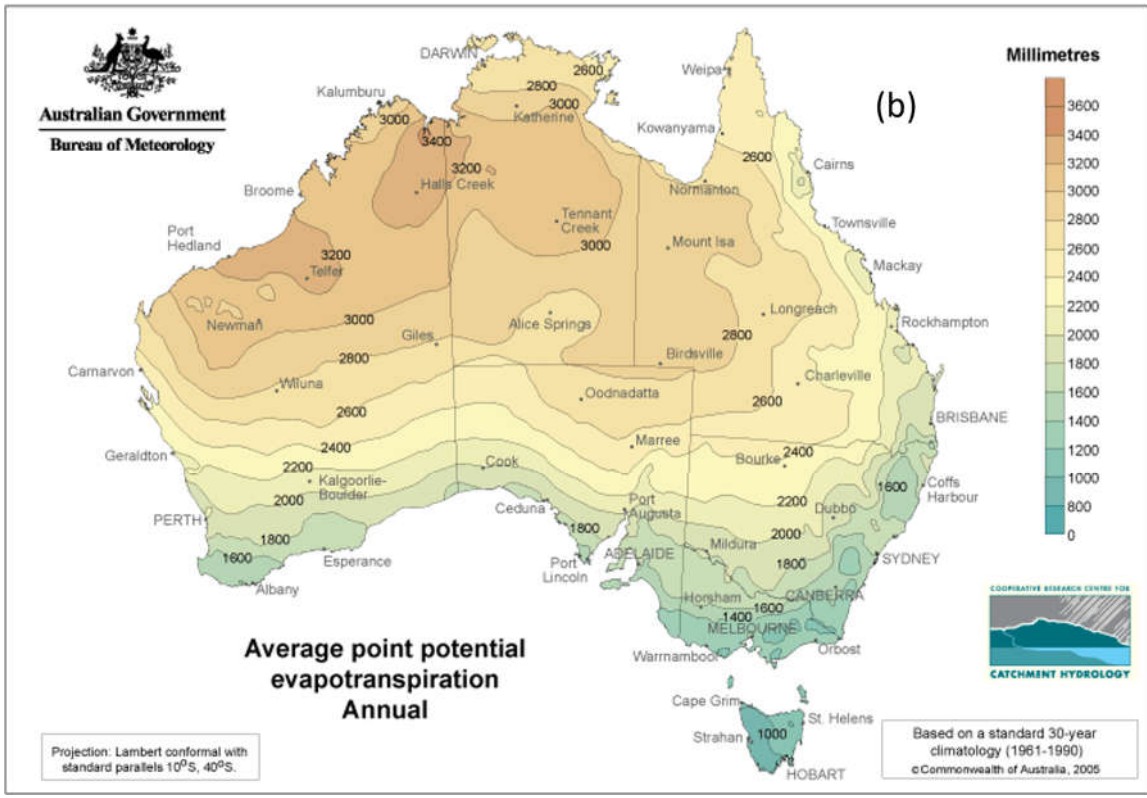

**Figure A2.** The spatial distribution of annual rainfall and evaporation across Australia: (**a**) mean annual rainfall; (**b**) mean annual potential evapotranspiration (PET). The annual mean is calculated using 30 years of gridded data between 1981 and 2010 for rainfall and 1975 and 2005 data for pan evaporation (Bureau of Meteorology: http://www.bom.gov.au/climate/maps/averages/, accessed on 24 May 2023).

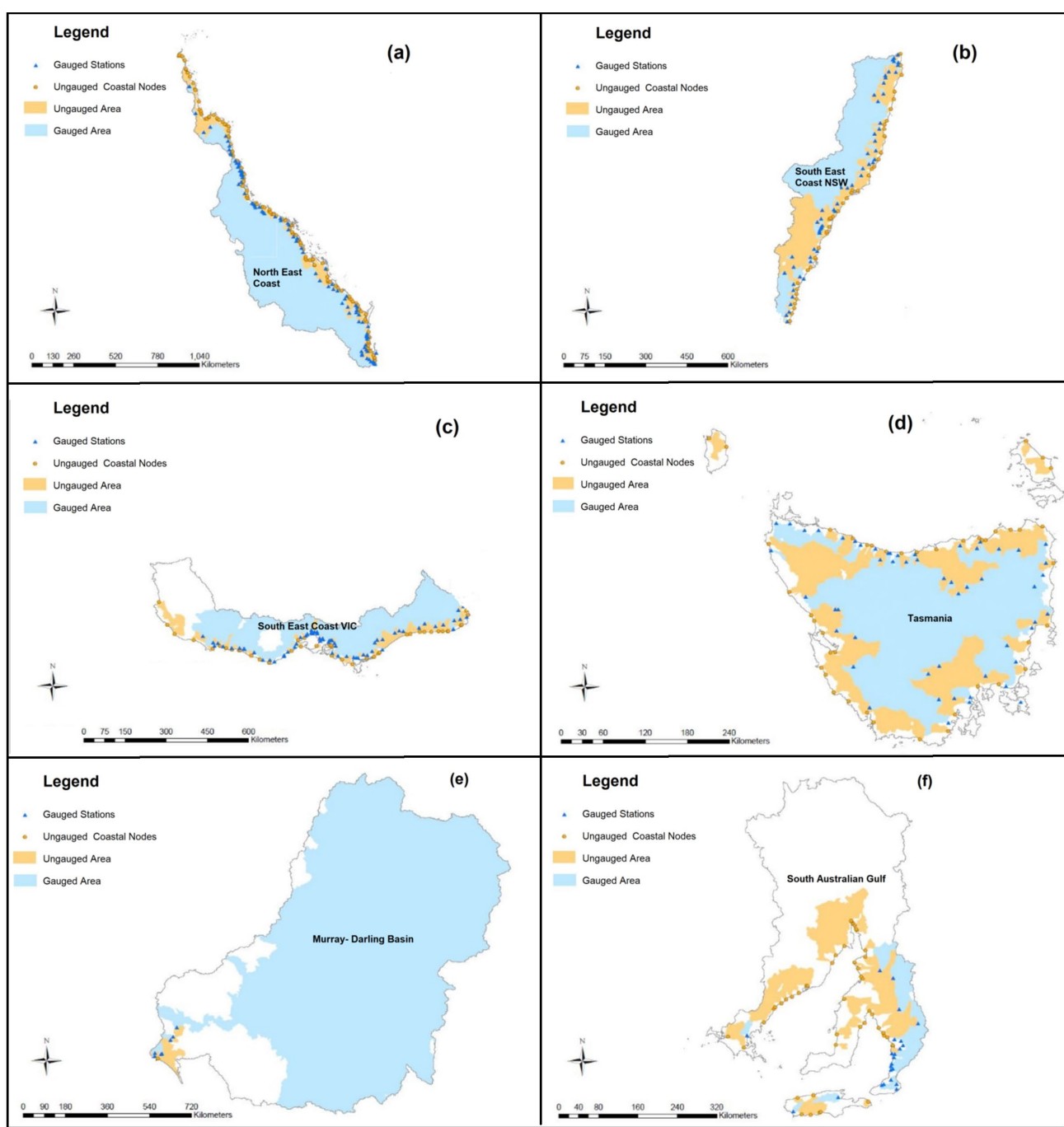

**Figure A3.** *Cont.*

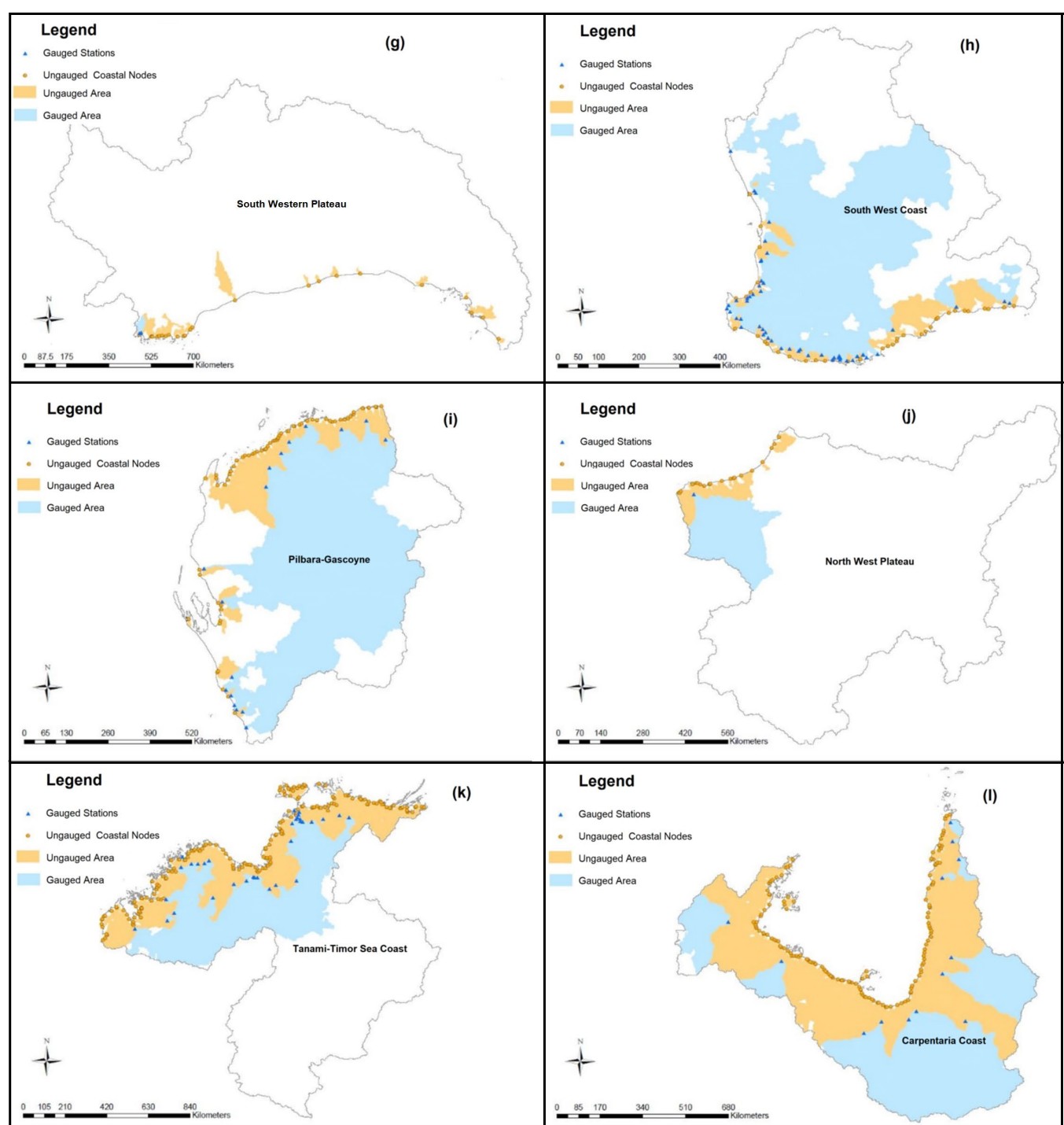

**Figure A3.** Ungauged catchments and nodes: (**a**) Northeast Coast; (**b**) Southeast Coast NSW; (**c**) Southeast Coast Vic; (**d**) Tasmania; (**e**) Murray–Darling Basin; (**f**) South Australian Gulf; (**g**) Southwestern Plateau; (**h**) Southwest Coast; (**i**) Pilbara–Gascoyne; (**j**) Northwestern Plateau; (**k**) Tanami–Timor Sea Coast; and (**l**) Carpentaria Coast.

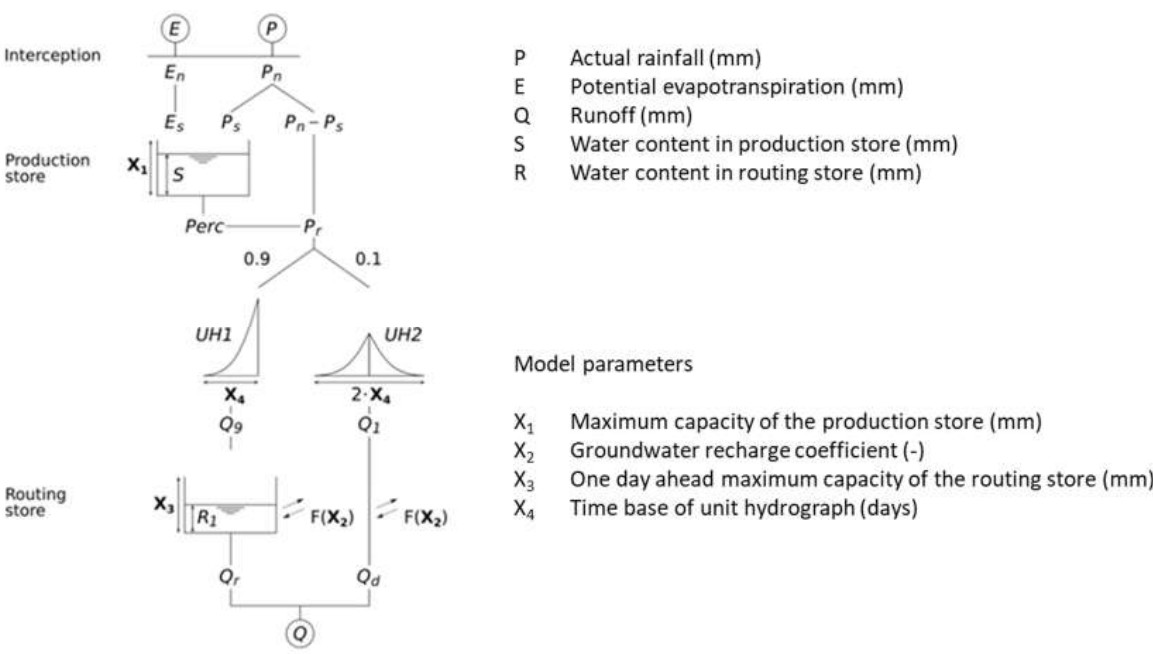

**Figure A4.** Description of GR4J model—conceptual representation of a sub-catchment with a river network and sub-areas [16].

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
