# Peer review of "Simulation of Gauged and Ungauged Streamflow of Coastal Catchments across Australia"

_water, doi:10.3390/w16040527_

Round 1

Reviewer 1 Report

Comments and Suggestions for Authors

Understanding the dynamics of coastal processes and the improvement of marine modelling systems and tools hinge significantly on precise streamflow estimation, particularly pertinent in the context of ungauged catchments. The purpose of this study was employing the GR4J lumped conceptual model to simulate streamflow across Australias coastal catchments, both gauged and ungauged, thereby creating a comprehensive nationwide dataset. However, the manuscript exhibits some areas for improvement in terms of structural coherence and linguistic expression, particularly with respect to paragraph organization and layout. Therefore, the reviewer recommends that this manuscript should be major revised.

(1) There appears to be a discrepancy in the section numbering. Notably, the ‘Methodology’ section is designated as the third section (3. Methodology), while logically, it should be the fourth section, according to the manuscript’s structure.It is suggested to check the section numbering in the paper carefully to ensure their coherence and accuracy.

(2) A discrepancy is observed in the manuscript’s section organization. The title of Section 3 is ‘Data Quality Control and Gap Filling’, however, only ‘Data Quality Control’ is discussed in this section as 3.1. The ‘Gap Filling’ part is instead discussed in Section 4 ‘Methodology’ as 4.2. This could potentially cause confusion for the readers. It would be more coherent and logical to move the ‘Gap Filling’ subsection from Section 4 to Section 3, making it a new subsection 3.2 under ‘Data Quality Control and Gap Filling’. 

(3) Section 4.1 reveals a significant omission in the employed methodology. Key components of water balance modelling such as water storage inflow, release, spill, draw, diversion, return flow, and evaporation have been excluded for the sake of simplicity. While the need for simplicity could be understood, the exclusion of these factors could potentially impact the accuracy and comprehensiveness of the model’s application. It would be recommended to consider the inclusion of these factors in their model or provide a more detailed justification for their exclusion.

(4) Section 4.4.1 appears to lack a clear justification for the statement concerning the relaxation of constraints for daily streamflow statistics. The paper state that ‘some of the constraints for the recommended statistics can be relaxed for daily streamflow’, however, it does not provide a clear rationale or evidence to support this claim. It would be beneficial to include a more detailed rationale as to why an NSE value lower than 0.5 for daily streamflow is still considered satisfactory, and how this directly leads to the conclusion that some constraints for the recommended statistics can be relaxed for daily streamflow. 

(5) The manuscript currently titles Section 5 as ‘Results and Discussion’. However, it is generally observed in academic writing that ‘Results’ and ‘Discussion’ are typically presented as two separate sections. This separation allows for a clearer distinction between the presentation of findings (Results) and their interpretation (Discussion). Therefore, it is recommended to introduce a new seventh section titled Discussion, and then move the ‘Future Research’ subsection (currently 5.6) from the fifth section to this newly created seventh section. This would allow for a more comprehensive discussion of the study’s findings and their implications.

(6) In the manuscript’s Section 6, titled ‘Summary and Conclusion’, the second paragraph contains a statement that appears to delve into the methodology, rather than providing a summary or conclusion of the research findings. The placement of the statement in the 'Summary and Conclusion' section could potentially confuse readers, as it seems to delve into the methods used in the study rather than summarizing the results or drawing conclusions. Therefore, it is suggested that this statement be moved to the 'Discussion' section (Section 7). This would allow for a more comprehensive discussion of the study’s methodology.

(7) It is observed that the readability of the figures could be improved. Specifically, the use of a blue solid line and a grey dashed line may not provide sufficient contrast for clear differentiation in Fig. 5. It is suggested to change the grey dashed line to a red solid line to enhance the distinction between the two. Additionally, in Fig 10, the box plots for each drainage division group are closely spaced, potentially hindering readability. It is recommended to adjust the spacing between the box plots of each drainage division group.

Reviewer 2 Report

Comments and Suggestions for Authors

The manuscript has an interesting topic that may attract attention on hydrologic modeling of gauged/ungauged basins of Australia. There seem to be a lot of work put in especially on input data preparation, gap filling of streamflow measurements, ungauged basin categorization and result compilation. On the other hand, there are several questions/drawbacks/suggestions/misprints marked on the reviewed manuscript where some of the more important points are reiterated below:

1.       Please recheck keyword selection, some of them do not seem to summarize the study clearly.

2.       Figure A3 (Appendix) needs a much better resolution. It could not be read in this form.

3.       It is stated that streamflow for each catchment is prepared from 1993 onward, but it is not clear how many years of data is used for modeling, at least on the average. A minimum of 5 years is set as a criterion, but it would be valuable to give an average modeling period.

4.       Runoff gap filling is explained in detail, but it would be good to know how much of the data is filled on the average.

5.       A little more clear information should be provided on regulating structures within the basins. It is mentioned that for simplicity, water balance modeling is not included. Does this mean modeling is done on regulated flow?

6.       Location of Table 3 is not appropriate if it is to include some results.

7.       In the results section when findings are compared with similar modeling approaches, a difference of around 20-35% is summarized as fairly good. Please shortly comment on this difference.

8.       When the results of the study are compared with similar previous research (especially references 30 and 64 where an hourly model GR4H is seen to be utilized along with catchment delineation including a channel routing scheme) a different modeling approach is undertaken. It should be clearly explained why a simpler version of the model (GR4J) is preferred in this study without routing. Because this is one of the main explanations for the deteriorating results.

Reviewer 3 Report

Comments and Suggestions for Authors

Review Report

A continuous daily streamflow time series of gauged and ungauged catchments has been developed for Australia. It is an applied research highly useful for the Australian community. A huge work has been done on the basis of a large amount of data. Genie Rural 4 parameter Journalier (GR4J) lumped conceptual model has been applied to predict stream flows from Australian catchments. The research work is useful but its originality needs to be highlighted. Further, the following comments may kindly be incorporated for improvement of the manuscript.

1                     Typo and English              

Line No 12,        Kindly change “majority of catchments”      to “majority of its catchments”

Line No 12, 17,    Kindly change “The mean annual rainfall and potential evapotranspiration (PET) and its spatial and temporal distribution vary significantly from one drainage division to another.”                      to             “The spatial and temporal distribution of mean annual rainfall and potential evapotranspiration (PET) vary significantly from one drainage division to another.”

Line 233, 234, Daily rainfall and potential evapotranspiration data are available at a 5 km by 5 km grid across Australia.

The PET data are calculated using the Priestley-Taylor evaporation equation.

Kindly clarify whether PET data was available or calculated. If it is available then why it is calculated?

Line 242    “for model GR4J model calibration”           delete one “model”

Line 246   Kindly explain what is “model warm-up”

2) Some explanations required

Line 249-252    Kindly describe how many data values were missing, erroneous, and constant, for which the gap-filling technique was used.

Kindly be consistent in using various words like “data set”  and   “dataset”

Kindly define what is meant by “runoff coefficient”

Kindly define what is meant by “GL”, A unit used for the stream flow

Kindly define every abbreviation before its first use.

Kindly provide values of calibrated parameters for various catchments.

Kindly provide a separate section for the validation of model results.

Kindly explain the methodology used in “the Shuffle Complex Evolution-University of Arizona (SCE-UA) algorithm”

Kindly explain that the observed stream flows shown in Figure 3 are related to which catchment area.

Kindly add the title of the x-axis in Figure 5.

Comments on the Quality of English Language

Minor editing of English language required

Round 2

Reviewer 1 Report

Comments and Suggestions for Authors

Understanding the dynamics of coastal processes and the improvement of marine modelling systems and tools hinge significantly on precise streamflow estimation, particularly pertinent in the context of ungauged catchments. The purpose of this study was employing the GR4J lumped conceptual model to simulate streamflow across Australias coastal catchments, both gauged and ungauged, thereby creating a comprehensive nationwide dataset. Although the manuscript exhibits some areas for improvement in terms of structural coherence and linguistic expression,  the reviewer recommends that it would be generally accepted.

(1)Although Section 4.1 has explained why key components of water balance modelling such as water storage inflow, release, spill, draw, diversion, return flow, and evaporation have been excluded, it would be beneficial to mention in the future work section that considering these factors could enhance the accuracy and reliability of the model predictions.

(2)In Section 4.3.1, the assertion ‘but in our study, we kept the performance identical to the monthly streamflow’ necessitates additional elucidation. Providing more context or details to clarify this point would be advantageous.

(3)The overall layout and language usage of the paper need to be carefully considered and refined. It would be recommended to review the manuscript thoroughly to ensure clarity and coherence. For example, the large blank space between Section 5.4 ‘Performance Evaluation - Ungauged Catchments’ and Figure 9 should be adjusted as it impacts the overall presentation of the paper.

Reviewer 2 Report

Comments and Suggestions for Authors

Dear Authors,

A lot of editing and corrections have been performed to improve the manuscript. One minor confusion is detected. Fig 1 and Fig A3 display one of the regions as South Western Plateau (SWP) whereas Table 4 and 5 describe it as South Australian Plateau (SAP). Please make a note.
